# Increase in *Akkermansiaceae* in Gut Microbiota of Prostate Cancer-Bearing Mice

**DOI:** 10.3390/ijms22179626

**Published:** 2021-09-06

**Authors:** Pin-Yu Huang, Yu-Chih Yang, Chun-I Wang, Pei-Wen Hsiao, Hsin-I Chiang, Ting-Wen Chen

**Affiliations:** 1Institute of Molecular Medicine and Bioengineering, National Yang Ming Chiao Tung University, Hsinchu 300, Taiwan; fdsa123j@gmail.com; 2Agricultural Biotechnology Research Center, Academia Sinica, Nangang District, Taipei City 115, Taiwan; ycyang12@gate.sinica.edu.tw (Y.-C.Y.); pwhsiao@gate.sinica.edu.tw (P.-W.H.); 3Radiation Biology Research Center, Institute for Radiological Research, Chang Gung University/Chang Gung Memorial Hospital, Linkou 333, Taiwan; yeewang0330@gmail.com; 4Department of Animal Science, National Chung Hsing University, Taichung 402, Taiwan; 5The iEGG and Animal Biotechnology Center, National Chung Hsing University, Taichung 402, Taiwan; 6Institute of Bioinformatics and Systems Biology, National Yang Ming Chiao Tung University, Hsinchu 300, Taiwan; 7Department of Biological Science and Technology, National Yang Ming Chiao Tung University, Hsinchu 300, Taiwan; 8Center for Intelligent Drug Systems and Smart Bio-Devices (IDS2B), National Yang Ming Chiao Tung University, Hsinchu 300, Taiwan

**Keywords:** gut–cancer axes, 16S rRNA, time-series, amplicon sequencing, microbiota comparison, hormone

## Abstract

Gut microbiota are reported to be associated with many diseases, including cancers. Several bacterial taxa have been shown to be associated with cancer development or response to treatment. However, longitudinal microbiota alterations during the development of cancers are relatively unexplored. To better understand how microbiota changes, we profiled the gut microbiota composition from prostate cancer-bearing mice and control mice at five different time points. Distinct gut microbiota differences were found between cancer-bearing mice and control mice. *Akkermansiaceae* was found to be significantly higher in the first three weeks in cancer-bearing mice, which implies its role in the early stage of cancer colonization. We also found that *Bifidobacteriaceae* and *Enterococcaceae* were more abundant in the second and last sampling week, respectively. The increments of *Akkermansiaceae*, *Bifidobacteriaceae* and *Enterococcaceae* were previously found to be associated with responses to immunotherapy, which suggests links between these bacteria families and cancers. Additionally, our function analysis showed that the bacterial taxa carrying steroid biosynthesis and butirosin and neomycin biosynthesis were increased, whereas those carrying naphthalene degradation decreased in cancer-bearing mice. Our work identified the bacteria taxa altered during prostate cancer progression and provided a resource of longitudinal microbiota profiles during cancer development in a mouse model.

## 1. Introduction

Prostate cancer is the second leading cause of death in the male population [1]. Although the pathogenesis of prostate cancer is strongly linked to its dependence on androgen hormones, the discovery of other risk factors involved in cancer development, progression, or resistance to therapies is of particular interest. The tumor microenvironment mediates tumor growth, cancer progression, invasion, and therapeutic resistance [2]. A variety of inflammatory cell types have been proposed to mediate changes in the microenvironment of prostate cancer, among them are macrophages and T cells, particularly IL-17-secreting Th17 cells [3,4,5]. On the tests of prostate and colon cancer in animal models, these cell types were found to promote carcinogenesis or tumor progression via STAT3 activation [3,6].

Many factors have been proposed as triggers of prostate inflammation, including viruses, bacteria, hormones, urinary reflux, and diet [7]. Human bodies contain trillions of bacteria that outnumber normal cells by up to 10 to 1, where the largest numbers of bacteria actually exist in the gut. The microbiota is composed of commensal bacteria and other microorganisms that lived on the epithelial barriers of the host [8]. The most extensively studied microbiome is the gut microbiome that performs some vital functions such as hydrolysis of dietary compounds, vitamin production, control of pathogen colonization, and protection from systemic infection [9]. Microbiota and host animals form a complex symbiotic relationship, which confers benefits to the host’s health and modulates host metabolism, inflammation, and immunity. Since information on the normal composition and functions of the microbiota has already been gained in the gastrointestinal and genitourinary systems, attention is now focused on the alteration in its composition when specific diseases arise, including cancer [10].

Increasing evidence suggests that the microbiota is involved in the tumor microenvironment and may play a role in the tumorigenesis process. Intestinal microbiota may have both an anti-tumoral [11,12,13] and a pro-tumoral effect [14,15,16,17]. The microbiota can influence cancer development and progression because of its potential to modulate inflammation and genomic stability of host cells [6]. The dysregulation of bacterial sensing and homeostasis between host and microbiome, also known as dysbiosis, may aggravate the tumor-elicited systemic inflammation and the oncogenesis of various types of cancer [18].

Where stomach cancer, colorectal carcinoma, and gallbladder carcinoma were associated with the local effects of the gastrointestinal microbiota, thymic lymphoma, mammary carcinoma, pancreatic cancer, prostate cancer, and ovarian cancer are found to be associated with the systemic effects of microbiota through the translocation of bacteria, bacterial toxins, cytokines, endocrines, or the migration of innate and adaptive immune cells that promote tumor oncogenesis from initiation to progression [8]. Likewise, recently, the microbiome has been suggested to modulate the efficacy of anti-cancer treatments such as chemotherapy, radiotherapy, immunotherapy, and hormonal therapy. Many anti-cancer therapy-promoting effects are mediated by immune system priming and by augmenting the immune response against tumor cells assisted by the effect of the microbiome [9].

The role of the microbiota in promoting the status of chronic inflammation and its implication in prostate cancer development has been suggested [19]. The discovery of a urinary microbiome composed of many different microorganisms supports this hypothesis because the prostate is exposed to many inflammatory stimuli derived from the bacteria in the surrounding environment [19]. Although it is difficult to confirm in human patients, Poutahidis et al. found that in a B6 Apc^Min/+^ mutant mouse model, the infection of intestinal *Helicobacter hepaticus* can trigger systemic elevations in pro-inflammatory cytokines (Eotaxin, IL-3, TNF-αalpha, and IL-1α) and enhance the prostatic intraepithelial neoplasia and micro-invasive carcinoma [20]. Sfanos and colleagues sequenced 16S rDNA amplicon of fecal microbiota from 30 subjects including healthy male volunteers and men with biochemically recurrent, localized, and metastatic prostate cancer [20]. The results showed a higher abundance of *Akkermansia muciniphila* and *Ruminococcaceae* spp. in the gastrointestinal microbiota of men being treated with oral androgen receptor axis-targeted therapies. This implied an intricate connection that may exist between microbiota and hormones in those cancers deeply influenced by hormones levels, such as prostate cancer, that recognize androgen hormones for guiding its development, treatment, and resistance to therapies [21].

Recent progress suggests that metastasis-initiating cells (MIC) with stem-like and immune-evasive properties can sustain through cancer therapy and bolster metastasis progress, which includes three distinct phases of dissemination, dormancy, and colonization [22]. MICs shuttle between quiescent and proliferative states during dormancy, at which stage the proliferative cells are continually eliminated by environmental fitness and niche-specific or systemic immune defenses. The outbreak of seeded MIC from metastatic dormancy relies on the dynamic equilibrium of various barriers and adaptations and can be triggered by oxidative stress, nutrients, and immune suppression signals [23]. Although the cause–effect relationship between gut microbiota and disease progression of cancer metastasis remains unclear, the dysbiosis induced by the gut microbiome mentioned above may promote the tumor-instigated inflammation and form potential tripartite interactions among the host immune system, the disseminated tumor, and the gut microbiome.

To delineate the role of the gut microbiome in the tripartite interaction, we studied the in vivo metastatic progression of human prostate cancer in the NOD-SCID mouse model and compared the 16S rRNA marker gene of gut microbiota between tumor-bearing and control (healthy) mice accordingly with a time-series progression of tumor development. The goal of this study is to identify potential marker bacterial taxa correlated with prostate cancer and characterize microbiome composition as an independent variable of metastasis in animals.

## 2. Results

To understand the alteration of gut microbiota in prostate cancer-bearing mice, fecal samples were collected from non-obese diabetic severe combined immunodeficient (NOD-SCID) mice. The injected tumor cells can be detected in the lung at the 4th week with IVIS imaging, hence we collected fecal samples on the 4th, 5th, 6th, 7th, and 8th week after seeding of prostate cancer cells (Figure 1). For each time point, we collected samples from three control tumor-free mice (designated ‘TF’) and three cancer-bearing mice (designated ‘T’). The microbiota profiles were established based on 16S rRNA amplicon sequencing data with QIIME2 (Appendix A Table A1). A total of 976 amplicon sequence variants (ASVs) and 235 taxa were identified in 30 samples.

### 2.1. Longitudinal Microbiota Profiles in Cancer-Bearing (T) and Tumor-Free (TF) Mice

In all the samples, the most abundant phyla were *Bacteroidetes* followed by *Firmicutes*, which is consistent with previous mouse gut microbiota studies (Figure 2a) [24]. Even though the phylum-level distributions were similar between T and TF groups, there were more *Verrucomicrobia* in T compared with TF, especially from the 4th to 6th week. The distributions are more diverse in the lower taxonomy levels. At the genus level, the top five most abundant taxa bar plots showed the relative abundance varied widely across the samples (Figure 2b). It is worth mentioning that the genus *Akkermansia*, which belongs to the phylum *Verrucomicrobia*, showed a higher abundance in group T.

### 2.2. Distinct Microbiota Compositions in T and TF Groups

We examined whether there was a difference between the longitudinal microbiota profiles in T and TF groups. We compared the alpha diversity and beta diversity between these two groups and between samples collected at different time points. We found significant differences in microbiota composition between the two groups. We also identified a significantly different abundance of families, *Akkermansiacease* and *Enterococcaceae*, in T and TF groups.

#### 2.2.1. No Significant Difference in Alpha Diversity Indexes between T and TF Groups

To explore the biodiversity within samples, we first compared four alpha diversity indexes, richness, Shannon index, evenness, and Faith PD from the 4th to 8th week. The alpha diversity was reported to be lower in the gut microbiota of patients diagnosed with prostate cancer [20]. However, we did not observe a statistically significant difference (Figure 3), which may result from the fact that that only three fecal samples in each group at each time point were examined. We further compared the alpha diversity indexes between T and TF groups, without separating them into different time points, and still found no significant difference.

#### 2.2.2. Separation of T and TF Groups in Beta Diversity

We found a significant difference between T and TF groups based on the beta diversity matrices, which suggests their microbiota compositions are significantly different. Principal coordinates analysis (PCoA) analysis based on Jaccard and Bray–Curtis dissimilarity matrices both showed that samples from the T group were separated from the TF group (Figure 4a,b). Notably, we found that samples collected at the same time also showed a tendency to cluster together. Jaccard and Bray–Curtis distance clearly distinguished the two groups. We also examined the dissimilarity distance within and between T and TF groups and found significantly higher dissimilarity distance between groups than within groups (Figure 4c,d).

We further examined whether this kind of dissimilarity was valid at all time points and whether there was a clear separation between T and TF groups (Figure 5a,c). Even though we constantly detected higher intergroup divergence than intragroup divergence, we obtained 14 significant adjusted *p* values out of 20 comparisons from all five time points (Figure 5b,d). Not all the comparisons were significant as expected, since we only had three samples in each group for each time point. Our results suggested that after the prostate cancer cells were seeded in mice, the gut microbiota may start to diverge and become meaningfully different as early as the 4th week.

### 2.3. Detection of Different Abundant Bacteria Family between T and TF Groups

To get a clear picture of alteration in gut microbiota, we investigated the different abundant taxa at each sampling time point with ANCOM and LEfSe [25,26]. The increase in *Akkermansiaceae* in the T group was supported by both tools. The significant increases in *the Akkermansiaceae* family in the T group at the 4th and 6th week were supported by ANCOM (Figure 6a,b). In LEfSe analysis, we found that the *Akkermansiaceae* family had significantly higher abundance in the T group at the 4th, 5th and 6th weeks with LDA scores higher than 4. Interestingly, we found the most abundant *Akkermansiaceae* in the 4th week in the T group and the abundance gradually decreased to the level of the TF group (Figure 6e). The average percentages of *Akkermansiaceae* were 19.7%, 8.3%, and 6.5% for the 4th, 5th, and 6th weeks, respectively. In addition to *Akkermansiaceae,* we also found a significant increase of *Bifidobacteriaceae* and *Enterococcaceae* in the 5th and 8th week (Figure 6c,d) with moderately enriched LDA scores (>3), respectively. Even though the relative abundance of *Enterococcaceae* was low (~0.5%), it was constantly detected in all three fecal samples taken at the 8th week (Figure 6f).

### 2.4. Correlation between the Abundance of Bacteria Families

We also explored the correlations between the abundance of bacteria families. We found three significantly positively correlated pairs, *Bifidobacteriaceae* and *Lactobacillaceae*; *Muribaculaceae* and *Akkermansiaceae*; *Saccharimonadaceae* and *Lachnospiraceae*. Among them, both *Bifidobacteriaceae* and *Lactobacillaceae* are well-recognized probiotic bacteria and their abundancies were frequently found positively correlated with each other [27]. Similar to *Akkermansiaceae*, *Muribaculaceae* was also found moderately enriched in group T in LEfSe analysis at the 5th, 6th, and 8th week with LDA score > 3. On the other hand, *Lactobacillaceae* showed significant negative correlations with *Ruminococcaceae* and *Lachnospiraceae*. *Bifdobacteriaceae* showed significant negative correlations with *Bacteroidaceae* and *Lachnospitaceae*. Additionally, *Barnesiellaceae* was found to be statistically significantly negatively correlated with *Akkermansiaceae* and *Lachnospiraceae* (Figure 7).

### 2.5. Function Prediction for Microbiota in T and TF Groups

The metabolic pathways for microbiota profiles from the two groups predicted with PICRUSt2 disclose three distinct KEGG pathways, steroid biosynthesis, butirosin and neomycin biosynthesis, and naphthalene degradation (Figure 8) [28,29]. Previous studies have shown that gut microbiota metabolizes and catabolizes sex hormones, affecting the levels of the hormones or their precursors in the host [30,31,32]. The higher abundance of steroid biosynthesis pathways suggests abnormal plasma levels of sex hormones, including androgen, which has been implicated in the development of the prostate gland and prostate cancer [33,34]. Hence, the enriched steroid biosynthesis pathway in the T group is especially noteworthy. The other two altered pathways, butirosin and neomycin biosynthesis and naphthalene degradation, have also previously been shown to be altered in gastric, bladder, colorectal, and ovarian cancers [35,36,37,38].

## 3. Discussion

The increased abundance of *Akkermansiaceae* in prostate cancer-bearing mice between the 4th and 6th week suggests a link between *Akkermansiaceae* and the early stage of injected cancer cell colonization. However, *A. muciniphila,* the most well-studied member of *Akkermansiaceae,* has probiotic properties and is usually found to be inversely related to human diseases such as obesity, diabetes, inflammation, and metabolic disorders [39,40,41,42]. Nevertheless, several studies suggest that an increase in *Akkermansia* in cancer patients may serve as a potential biomarker. For example, *Akkermansia* is more abundant in microbial extracellular vesicles from blood samples in pancreatic cancer patients [43]. Previous studies also suggested links between *A. muciniphila* and the development of colon cancer. Being a mucin-degradation bacterium [44], *A. muciniphila*, has the potential to induce colonic inflammation and cancer development [45]. Colonization of *A. muciniphila* is also significantly associated with increased tumor development in the mouse model and may promote tumorigenesis in human colon cancer [46,47,48].

It is worth noting that the abundance of *Akkermansiaceae* in cancer-bearing mice decreased gradually and then became indistinguishable from it in control mice. This kind of dynamic change of the abundance of *Akkermansiaceae* may be related to the process of prostate cancer colonization. The bacteria phylum *Verrucomicrobia*, to which *Akkermansia* belongs, appears to contribute to inflammation in mice gut microbiota [45,49], and its abundance increases along with chemokine/cytokine elevation in mucositis induced by the chemotherapeutic agent 5-Fu [50]. Additionally, extracellular vesicles from *A. muciniphila* were shown to increase the number of M1-like macrophages that produce inflammatory cytokines [51]. These dysregulations may contribute to shaping the metastatic niche for prostate cancer cells. Our findings of the dynamical changes during cancer colonization/development in gut microbiota also confirm that the time-series microbiota profiling strategy provides the possibility to capture the important transient states.

The *Enterococcaceae* family is altered in the microbiota in patients associated with many cancer types. For example, it was found to be more abundant in individuals with colon polyps, a risk factor for colorectal cancer [52]. Along with *Bifidobacteriaceae* and *Enterobacteriaceae*, *Enterococcaceae* was also found to be enriched with cholangiocarcinoma caused by Southeast Asian liver fluke (*Opisthorchis viverrini*) [53]. *Enterobacteriaceae*, *Pseudomonadaceae*, *Moraxellaceae*, and *Enterococcaceae* were found to be increased in pancreatic ductal adenocarcinoma tissues [54]. All these results suggest *Enterococcaceae* may play a role in cancer development.

Notably, *Akkermansiaceae*, *Bifidobacteriaceae*, and *Enterococcaceae* were found to be more abundant in cancer patients who responded to PD-1 therapy compared to non-responders [55,56,57]. Even though the detailed mechanism is still unknown, the associations between these taxa and immunotherapy responses suggest links between gut microbiota and immune evasion of cancers. Here, our findings of increased abundance of *Akkermansiaceae*, *Bifidobacteriaceae*, and *Enterococcaceae* at different time points may provide hints for the underlying mechanism and the time-series microbiota profiles during cancer colonization could provide a resource for the cancer immunotherapy research community.

The relative abundances of *Bifidobacteriaceae* and *Lactobacillaceae* were found to be positively correlated with each other. These two families contain many well-known probiotic species. For example, *Lactobacillus* and *Bifidobacterium* were found to be anti-tumorigenic by deactivating microbial enzymes involved in the production of carcinogens in colon cancer [58]. *Lactobacillus casei* was reported to secrete ferrichrome, a metabolite that can trigger JNK-mediated apoptosis of cancer cells in a mouse xenograft model [59]. *Bifidobacterium* species were also found to be correlated with PD-L1 treatment response. A combination of oral administration of *Bifidobacterium* and PD-L1 inhibitors activate T cells and block the growth of melanoma in mice [60]. A mixture of *Bifidobacterium longum* BB536 and *Lactobacillus johnsonii* La1 was also shown to decrease the concentration of pathogens and adjust intestinal immunity in colorectal cancer patients [61]. Recently, the FDA approved the first microbiome-based drug, SER-109, for treatment of recurrent *Clostridium difficile* infection (CDI) [62]. The potential of using a mixture of these identified correlated bacteria as a drug is worth further investigation.

The abundance of bacteria carrying genes involved in steroid biosynthesis in the T group is noteworthy. The requirement of androgen, a steroid hormone, for prostate cancer development has been known for half a century [34]. Studies in animal models have demonstrated the contribution of gut microbiota in shaping hormone levels in blood [63,64]. The gut microbiota was thought to be involved in prostate cancer development through the metabolism of these steroid hormones [65]. Moreover, androgen suppression therapy used in prostate cancer treatment increases the abundance of *A. muciniphila* and genes involved in steroid biosynthesis in gut microbiota [20]. Our findings support the role of the steroid biosynthesis pathway of gut microbiota in prostate cancer.

## 4. Materials and Methods

### 4.1. Experimental Design

All mice were maintained under the specific pathogen-free (SPF) conditions at Agricultural Biotechnology Research Center Academia Sinica. Six male NOD-SCID mice were kept with normal chow and autoclaved water and divided into TF and T groups (three each). T group mice were seeded with 3 × 10^5^ mCRPC 22Rv1-M4 [66] by tail vein injection at eight weeks old and monitored via an in vivo imaging system (IVIS). Gut microbiota was derived from the fecal samples of the mice. Three fecal samples were collected from each group every week from the 4th to the 8th week after the tumor inoculation.

### 4.2. Sample Collection and DNA Extraction

The fecal pellets were collected under SPF conditions and were kept in a centrifuge tube at −80 °C for DAN preparation. Within one week after collection, bacterial DNA was extracted from fecal samples using QIAamp^®^ Fast DNA Stool Mini Kit (Qiagen, Germantown, MD, USA) according to the manufacturer’s instruction.

### 4.3. 16S rRNA Gene Amplification and Sequencing

PCR reactions were used to amplify the targeted 16S rRNA region from DNA samples. The V3–V4 region of the bacterial 16S rRNA gene was amplified using 2.5 μL microbial DAN and forward primers 341F (5′-CCTACGGGNGGCWGCAG-3′) and 805R (5′-GACTACHVGGGTATCTAATCC-3′). PCR reactions were performed in 12.5 μL 2xKAPA HiFi HotStart ReadyMix (Sigma-Aldrich, St. Louis, MO, USA) at an annealing temperature of 55 °C for 25 cycles. The 16S V3–V4 amplicon was purified of free primers using AMPure XP beads (Beckman Coulter, Indianapolis, IN, USA) and eluted with 80% ethanol (EtOH). The PCR amplicons were sequenced by Illumina MiSeq paired-end reads (Illumina, San Diego, CA, USA).

### 4.4. Sequence Preprocessing and Taxonomy Classification

The sequence data were analyzed with QIIME 2 [67]. Raw paired-end 16S rRNA reads (V3–V4 region) imported into QIIME 2 were first trimmed with primer sequences (--p-front-f CCTACGGGNGGCWGCAG --p-front-r GACTACHVGGGTATCTAATCC). The sequences were then trimmed by length and base quality, and subsequently merged into consensus fragments using DADA2 with the parameter --p-trunc-len-f 283 --p-trunc-len-r 169 [68]. After dereplication, discarding the singletons, and removing chimera with DADA2, the remaining sequences were ASVs. Database SILVA release 132 was used for taxonomic classification of ASVs with QIIME2 [69]. Sequences that would theoretically be amplified by primer set 341F/805R were extracted from the SILVA dataset and used for taxonomic classifier training in QIIME2 (feature-classifier).

### 4.5. Diversity and Correlation Analysis

The ASVs were used for diversity analysis with the R package VEGAN or QIIME2 [67,70]. Specifically, rarefaction analysis and phylogenetic diversity were analyzed with QIIME2 and all the other diversity indexes were derived from package VEGAN. A non-parametric statistical method, the Wilcoxon rank-sum test (also called Mann–Whitney test), was used to compare alpha diversity indexes, richness diversity, Shannon diversity, Evenness diversity, and Faith’s phylogenetic diversity (Faith PD) between groups [71,72]. To evaluate the statistical significance of beta diversity distances, Jaccard distances and Bray–Curtis distances were first tested with permutational multivariate analysis of variance (PERMANOVA) [73]. The distance between each pair of samples from the same or different groups was then tested with the Wilcoxon rank-sum test [72]. Correlation coefficients between the abundance of bacteria taxa were conducted with the Spearman correlation. To correct for multiple tests, all the resulting *p* values were further adjusted by BH correction [74].

### 4.6. Detection of Different Abundant Bacteria Families and Function Prediction

LEfSe (linear discriminant analysis (LDA) effect size) [26] and ANCOM (analysis of the composition of microbiomes) [25] were applied to identify features (taxa) with different abundances among group TF and T at different time points. Function prediction from bacteria from 16S rRNA genes was conducted by PICRUSt2 (phylogenetic Investigation of Communities by Reconstruction of Unobserved States) [29] with KEGG (Kyoto Encyclopedia of Genes and Genomes) database [28]. STAMP v2 [75] (statistical analysis of taxonomic and functional profile) was used to identify the pathways with significantly different abundance. We compared the abundance of predicted functions between two groups by White’s non-parametric t-test and applied BH correction to get the final BH adjusted *p*-values [76].

## 5. Conclusions

This study provided time-series gut microbiota from control mice and prostate cancer-bearing mice. The microbiota profiles are distinct between the two groups at all sampling time points, i.e., 4th to 8th week after cancer inoculation. Our findings support the dynamic alteration of the mice’s gut microbiota after bearing human cancer cells. Our dataset also offers a resource for investigating the time-dependent alteration in gut microbiota for the cancer-bearing mouse model.

## Figures and Tables

**Figure 1 ijms-22-09626-f001:**
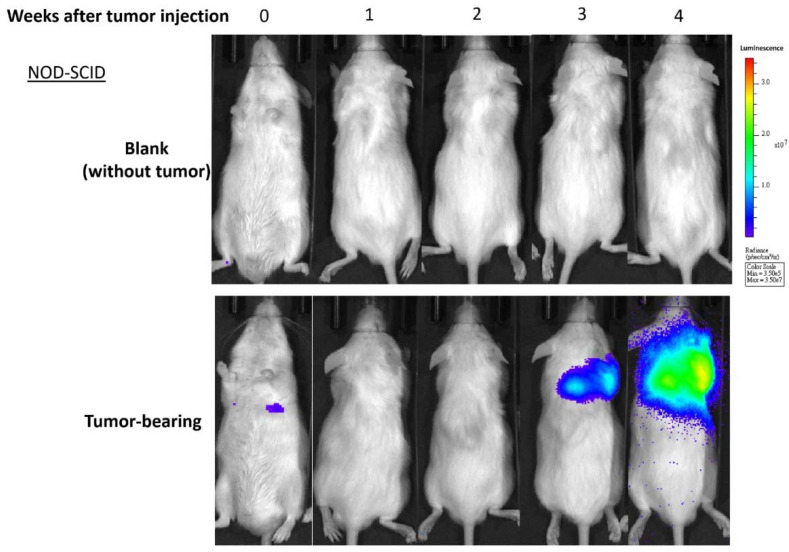
Tumor-specific bioluminescent images in representative mice. Groups of NOD-SCID mice (N = 5, age of eight weeks) were inoculated with 3 × 10^5^ cells of metastatic human prostate cancer 22Rv1-M4 cells vs. mock control (PBS). Based on the housekeeping expression of the luciferase 2 (Luc2) reporter gene in the tumor cells, in vivo bioluminescent imaging of tumors was captured and quantified using an IVIS Lumina XRMS System weekly as marked.

**Figure 2 ijms-22-09626-f002:**
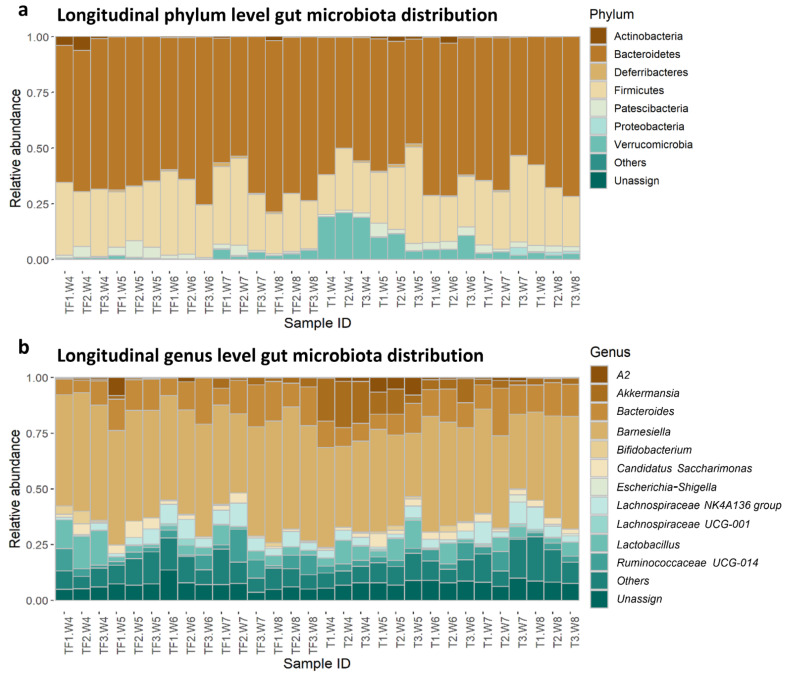
Relative abundance of identified taxa at (**a**) the phylum level and (**b**) the genus level. The top five most abundant taxa from all samples were plotted and all the other taxa were grouped as “others”. Taxa that cannot be assigned to a phylum or a genus with SILVA database were assigned to “Unassign”. *A2* genus represents a group of uncultured bacteria identified in mouse gut microbiota under *Lachnospiraceae*. The sample IDs consist of the group, mouse ID, and sampling time, e.g., TF1.W4 is the sample collected from mouse #1 in the TF group in the 4th week.

**Figure 3 ijms-22-09626-f003:**
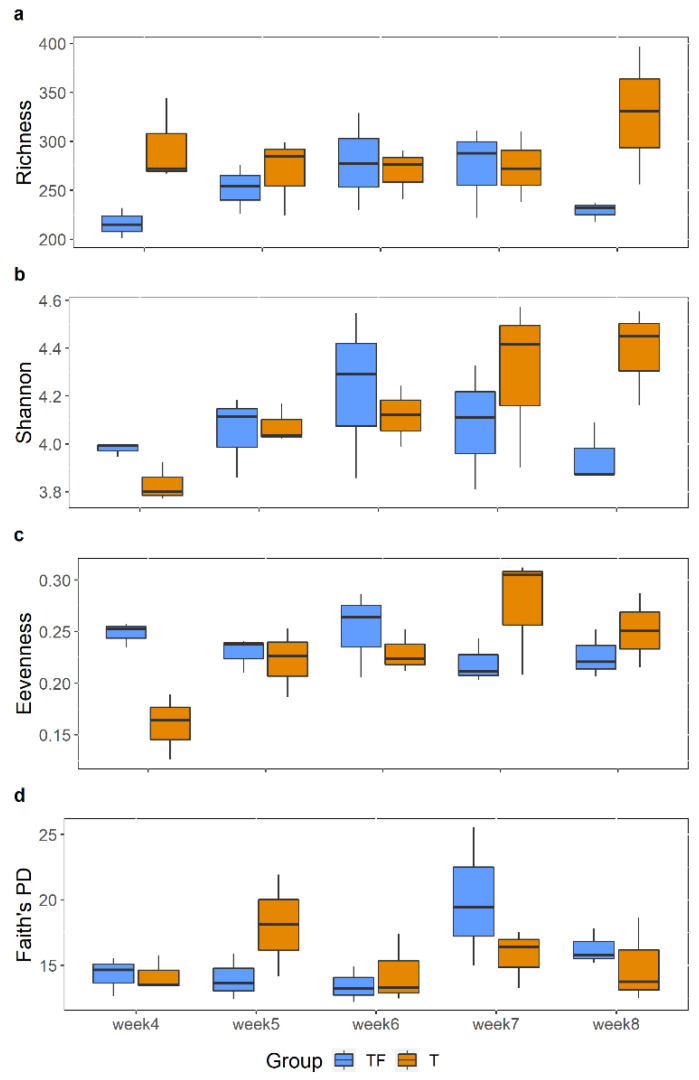
Comparisons of Alpha diversity indexes between TF and T groups at different time points. (**a**) Richness. (**b**) Shannon index. (**c**) Evenness. (**d**) Faith’s PD. Each box displays the alpha diversity distribution for three samples. There is no significant difference between TF and T at all five time points for all alpha diversity indexes (Wilcoxon rank-sum test).

**Figure 4 ijms-22-09626-f004:**
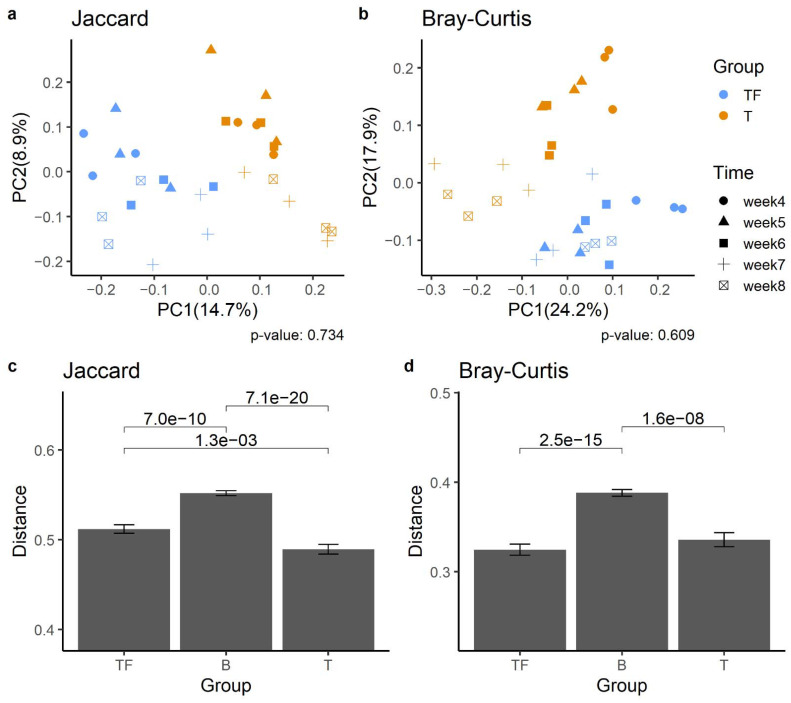
Beta diversity analysis of gut microbiota from T and TF groups. PCoA plots derived from (**a**) Jaccard distances and (**b**) Bray–Curtis distances show the separation of the two groups. The percentage of variation explained in PCoA plots is noted in parentheses at the x and y axes. The distribution of Jaccard (**c**) and Bray–Curtis (**d**) beta-diversity distances within and between T and TF groups reveal significantly different microbiota compositions between the two groups. The distances within TF and T groups are noted as TF and T, respectively, while the distance between TF and T groups is noted as B. *p* values derived from Wilcoxon rank-sum test are shown above the bars.

**Figure 5 ijms-22-09626-f005:**
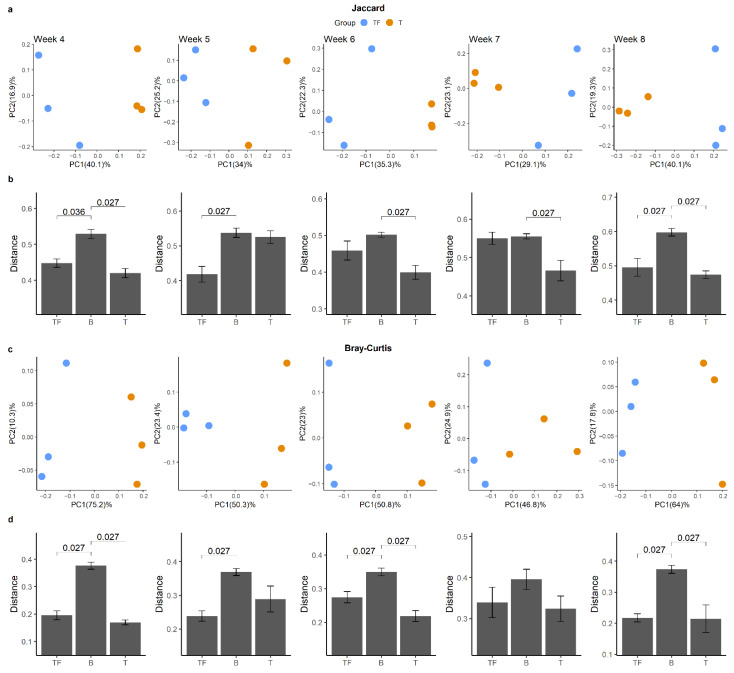
Beta diversity analysis for gut microbiota from two groups at different time points. PCoA plots derived from (**a**) Jaccard distance and (**c**) Bray–Curtis distance for the five time points show the separation between T and TF groups. The distribution of Jaccard (**b**) and Bray–Curtis (**d**) beta-diversity distances within and between T and TF groups reveals the difference between T and TF at different time points. The distances within TF and T groups are noted as TF and T, respectively, while the distance between TF and T groups is noted as B. *p* values derived from Wilcoxon rank-sum test are shown in the bar plots.

**Figure 6 ijms-22-09626-f006:**
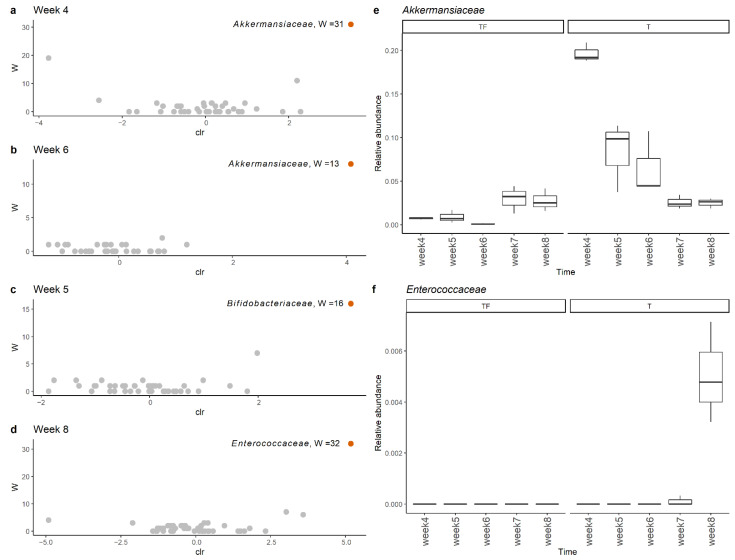
Comparison of microbiota profiles at the family level with ANCOM showed that *Akkermansiaceae* and *Enterocaceae* had significantly higher abundance in the T group at (**a**) week 4, (**b**) week 6, (**c**) week 5, and (**d**) week 8. The relative abundance of (**e**) *Akkermansiaceae* and (**f**) *Enterococcaceae* family at weeks 4, 5, 6, 7, and 8 in group T and group TF. *Akkermansiaceae* was more abundant at weeks 4, 5, and 6 while *Enterococcaceae* was more abundant at week 8 in group T.

**Figure 7 ijms-22-09626-f007:**
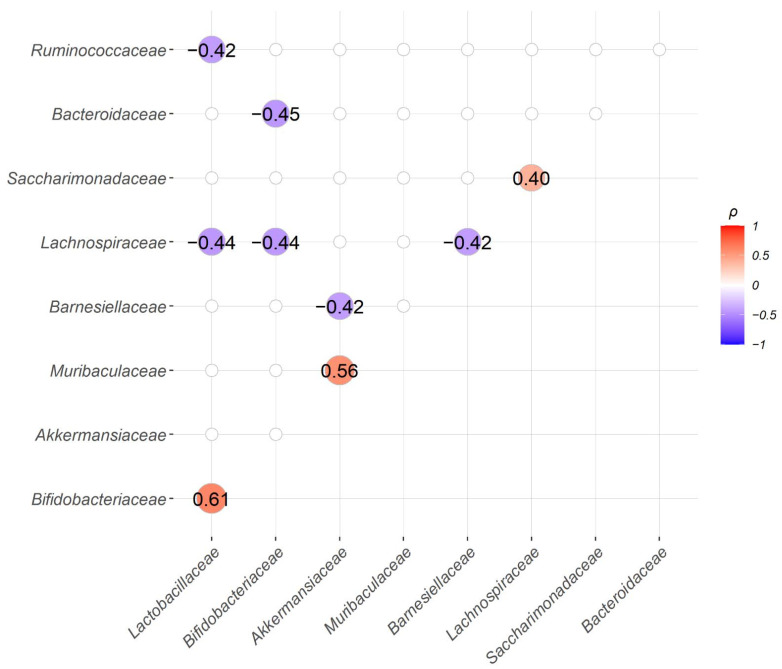
Correlation analysis between the abundance of the top 10 bacteria families, except for the “unassigned” group, showed three significant positive and six significant negative bacteria family pairs. Spearman’s correlation coefficients (ρ) for the abundance of paired bacteria families were calculated from all 30 samples and only family pairs with significant correlations are shown (BH adjusted *p*-value < 0.05).

**Figure 8 ijms-22-09626-f008:**
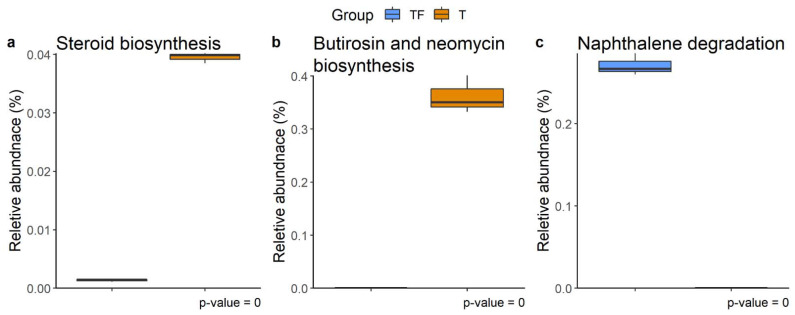
Different abundant KEGG metabolic pathways. (**a**) Steroid biosynthesis, (**b**) butirosin and neomycin biosynthesis, and (**c**) naphthalene degradation were found at week 4, week 7, and week 8, respectively. The relative abundance of function predicted by PICRUSt2 showed the different abundance of taxa predicted to have a specific function in T and TF groups. The BH adjusted *p* values from White’s non-parametric *t*-test are shown at the bottom of the box plots.

## Data Availability

All the sequencing data used are available in the SRA database under BioProject number PRJNA745267.

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
