# Peer review of "Increase in Akkermansiaceae in Gut Microbiota of Prostate Cancer-Bearing Mice"

_ijms, 2021, doi:10.3390/ijms22179626_

Round 1
Reviewer 1 Report
This work from Huang and colleagues focused on the gut microbiota trajectory in prostate cancer bearing-mice compared to healthy models. To my standpoint several limitations emerged: the lack of a baseline sample (i.e., before to induce tumor in mice), several errors in method section, the statistical methods in some parts are not proper described. Moreover, I know that NOD-SCID mice are a common model to study prostate cancer but I'm not quite sure this is the best model for analyzing gut microbiome profile because of the involvement of the gut microbial community in type1 diabetes development.
Author Response
Thanks for your suggestions. In the current study, we have designed the blank control (without injection of tumor cells to inoculate prostate cancer) to compare with the treatment groups. Because it is a paired comparison in which the results are monitored on a time-series basis, the blank control can serve as a good baseline for our purpose.
To improve the clarity of the statistical methods, we have revised the related contents in the method section as follows. Line 378: “To evaluate the statistical significance of beta diversity distances, Jaccard distances, and Bray-Curtis distances were first tested with permutational multivariate analysis of variance (PERMANOVA) [74]. The distance between each pair of samples from the same or different groups was then tested with Wilcoxon rank-sum test [73]. Correlation coefficients between the abundance of bacteria taxa were conducted with Spearman correlation. To correct for multiple tests, all the resulting p values were further adjusted by BH correction [75].”
Animals homozygous for the severe combined immunodeficiency (SCID) mutation have impaired T and B cell lymphocyte development without affecting the myelopoiesis. The Non-Obese Diabetic (NOD) mice congenic for the SCID mutation do not allow for the phenotypical expression of the autoimmune (type I) diabetes that characterizes the NOD background. Instead, the NOD background additionally results in a lack of circulating complements, defects of natural killer (NK) cells, and deficient maturation in macrophages and antigen-presenting cells (APCs) for the tumor immunosurveillance, thus becomes an optimal host to support human tumor cell growth and hematopoiesis (Chiu et al., 2002). In the current study, the NOD-SCID mice were chosen because they work consistently and reproducibly for the tumorigenesis of our metastatic prostate tumor model. Although NOD-SCID mice might not be the best model, they have been used for studying the relationships between intestinal microbiota and xenografted human pancreatic cancer as well (Thomas, et al., 2018). However, we agree that there is a crosstalk between the gut microbiota and immune state of the hosts, which was proven by observed nuances of gut microbiota between SCID and NOD/SCID mice (Zheng et al., 2019). To avoid the potential influence of immunodeficiency on gut microbiota in our study, a new animal trial using FVB/NJ mice (with the normal immune system) is now undergoing to validate our hypothesis.
Chiu P. P. L., Ivakine E., Mortin-Toth S., and Danska J. S. (2002) Susceptibility to Lymphoid Neoplasia in Immunodeficient Strains of Nonobese Diabetic Mice. Cancer Res. 15;62(20):5828-5834
Thomas, R. M., Gharaibeh, R. Z., Gauthier, J., Beveridge, M., Pope, J. L., Guijarro, M. V., Yu, Q., He, Z., Ohland, C., Newsome, R., Trevino, J., Hughes, S. J., Reinhard, M., Winglee, K., Fodor, A. A., Zajac-Kaye, M., & Jobin, C. (2018). Intestinal microbiota enhances pancreatic carcinogenesis in preclinical models. Carcinogenesis, 39(8), 1068–1078.
Zheng, S., Zhao, T., Yuan, S., Yang, L., Ding, J., Cui, L., & Xu, M. (2019). Immunodeficiency Promotes Adaptive Alterations of Host Gut Microbiome: An Observational Metagenomic Study in Mice. Frontiers in microbiology, 10, 2415.

Reviewer 2 Report
The submitted paper entitled „Increase of Akkermansiaceae in gut microbiota in prostate cancer-bearing mice” is a very interesting and original research article, planned and carried out with great care.
I have found a few minor suggestions. Enclosed is a reviewed manuscript with proposed changes and a few questions - I would be grateful if the authors could explain:
- what was the proof of concept of this study?
- Why do the authors place these results as an appendix and not in the main text?
Moreover, please improve the quality of all images, mainly Figure 1 a-b, Figure 3 a-f, as well as put ‘references’ in the right order in the manuscript – here are recommendations from MDPI, available online: https://www.mdpi.com/authors/references

Author Response
Thank you for your suggestions. The ‘concept’ in the introduction was referring to the discovery of systematic effects on prostate cancer tumorigenesis initiated by intestinal microbiota. However, to avoid confusion, this part of contents has been revised as follows: (Line 88) “Although it is difficult to confirm in human patients, Poutahidis, et al. found that in a B6 ApcMin/+ mutant mice model, the infection of intestinal Helicobacter hepaticus can trigger systemic elevations in pro-inflammatory cytokines (Eotaxin, IL-3, TNF-αalpha, and IL-1α) and enhance the prostatic intraepithelial neoplasia and micro-invasive carcinoma [20].”
Following your suggestion, we have changed the names of bacteria genera and families in all figures and figure legends into italic fonts. We also replaced all the figures with high-resolution and the high-resolution figures (300 dpi) for original Figure 1 (now Figure 2) and Figure 3 (now Figure 6), which are provided as individual TIFF files as well. The figures in the appendix were relocated to the main text as Figure 1, Figure 3, and Figure 5, but Table A1, the statistical summary of the sequencing data, was kept in Appendix. The reference section is presented right after “Conflicts of Interest”.

Reviewer 3 Report
The paper opens new perspectives about the role of Akkermansia in Gut Microbiota and it can permit to perform new study
Author Response
Thank you for the comments. We sincerely appreciate your efforts in reviewing our manuscript.
